# Socioeconomic deprivation and illness trajectory in the Scottish population after COVID-19 hospitalization

## Abstract

**Background** The associations between deprivation and illness trajectory after hospitalisation for coronavirus disease-19 (COVID-19) are uncertain.
**Methods** A prospective, multicentre cohort study was conducted on post-COVID-19 patients, enrolled either in-hospital or shortly post-discharge. Two evaluations were carried out: an initial assessment and a follow-up at 28–60 days post-discharge. The study encompassed research blood tests, patient-reported outcome measures, and multisystem imaging (including chest computed tomography (CT) with pulmonary and coronary angiography, cardiovascular and renal magnetic resonance imaging). Primary and secondary outcomes were analysed in relation to socioeconomic status, using the Scottish Index of Multiple Deprivation (SIMD). The EQ-5D-5L, Brief Illness Perception Questionnaire (BIPQ), Patient Health Questionnaire-4 (PHQ-4) for Anxiety and Depression, and the Duke Activity Status Index (DASI) were used to assess health status.
**Results** Of the 252 enrolled patients (mean age $55.0 \pm 12.0$ years; 40% female; 23% with diabetes), deprivation status was linked with increased BMI and diabetes prevalence. 186 (74%) returned for the follow-up. Within this group, findings indicated associations between deprivation and lung abnormalities ($p = 0.0085$), coronary artery disease ($p = 0.0128$), and renal inflammation ($p = 0.0421$). Furthermore, patients with higher deprivation exhibited worse scores in health-related quality of life (EQ-5D-5L, $p = 0.0084$), illness perception (BIPQ, $p = 0.0004$), anxiety and depression levels (PHQ-4, $p = 0.0038$), and diminished physical activity (DASI, $p = 0.002$). At the 3-month mark, those with greater deprivation showed a higher frequency of referrals to secondary care due to ongoing COVID-19 symptoms ($p = 0.0438$). However, clinical outcomes were not influenced by deprivation.
**Conclusions** In a post-hospital COVID-19 population, socioeconomic deprivation was associated with impaired health status and secondary care episodes. Deprivation influences illness trajectory after COVID-19.

### Plain language summary

In our study, we aimed to understand how socioeconomic factors impact recovery from COVID-19 following hospitalisation. We followed 252 patients, collecting health data and utilising advanced imaging techniques. We discovered that individuals from deprived areas experienced more severe health complications, reported worse quality of life, and required more specialist care. However, their clinical outcomes were not significantly different. This underscores that socioeconomic deprivation affects health recovery, underlining the need for tailored care for these individuals. Our findings emphasise the importance of considering socioeconomic factors in recovery plans post-COVID-19, potentially improving healthcare for those in deprived areas.

Socioeconomic factors influence health outcomes in pandemics due to influenza[1,2] and COVID-19[2–4]. Observational and population studies suggest that people from socially deprived areas have a greater risk of developing SARS-CoV-2 infection, have more severe acute symptoms, may be at greater risk of developing post-COVID conditions (long COVID) and are at higher risk of COVID-related mortality[5–7]. However, at the onset of the COVID-19 pandemic, clinical studies were deficient in prospective evaluations of disease pathogenesis and health status. They also selectively recalled patients, introducing selection bias. Consequently, the cause of this relationship remains unclear[8,9]. It is hypothesised that factors such as socioeconomically disadvantaged individuals having greater occupational exposure to COVID-19[10], reduced access to personal protective equipment (PPE)[11], greater multimorbidity[12], and more unhealthy lifestyle factors may contribute[13].

✉e-mail: colin.berry@glasgow.ac.uk

The Chief Scientist Office Cardiovascular and Pulmonary Imaging in SARS Coronavirus disease-19 (CISCO-19) study is a prospective, observational, multicentre, longitudinal, secondary care cohort study that assessed the time-course of multi-organ injury in post-hospital survivors of COVID-19 during convalescence and controls[14]. Adjudicated myocarditis persisting 28–60 days post-COVID-19 affected 1 in 8 (13%) patients, and the likelihood of myocarditis was associated with lower health-related quality of life, enhanced illness perception, enhanced depression score, lower physical activity and lower predicted maximal oxygen utilization (ml/kg/min). One in seven patients died or were rehospitalized and two in three patients had additional outpatient episodes of secondary care, considerably higher than controls[14].

In this prespecified analysis, we investigate the pathophysiological associations of socioeconomic status documented at baseline and disease trajectory in COVID-19.

This analysis reveals significant associations between socioeconomic deprivation and various health outcomes in patients post-COVID-19. Key findings include a higher prevalence of obstructive coronary artery disease, persistent lung abnormalities, and renal inflammation in patients from more deprived areas. Additionally, these individuals reported lower health-related quality of life, enhanced illness perception, and reduced physical function. Mental health disparities were also evident, with less improvement in depression scores among more deprived patients. These trends underscore the influence of socioeconomic factors on post-COVID-19 recovery, highlighting the need for targeted healthcare strategies to address these disparities.

## Methods
### Study design
This study involved a prospective, observational, multicenter, longitudinal, secondary care cohort design to assess the time course of multi-organ injury in survivors of COVID-19 during convalescence (ClinicalTrials.gov ID NCT04403607). The design, baseline characteristics, and primary outcome results of the study have been described[14,15]. Clinical information, a 12-lead digital ECG, blood and urine biomarkers, and patient-reported outcome measures were acquired at enrolment (visit 1) and again during convalescence, 28–60 days post-discharge (visit 2). Chest computed tomography (CT), including pulmonary and coronary angiography, and cardio-renal MRI, were acquired at the second visit.

The Scottish Index of Multiple Deprivation (SIMD) is a relative measure of deprivation across 6976 small areas within Scotland, UK (called data zones) based on domiciled post code reflecting seven factors (income, employment, education, health, access to services, crime, and housing) and categorized into general population quintiles[16]. Quintile 1 (Q1) represents the most socioeconomically deprived areas, and Q5 the least.

The EQ-5D-5L, Brief Illness Perception Questionnaire (BIPQ), Patient Health Questionnaire-4 (PHQ-4) for Anxiety and Depression, and the Duke Activity Status Index (DASI) were used at enrolment and 28–60 days to assess health status.

### Study setting
The study involved three hospitals in the West of Scotland (population 2.2 million) - the Queen Elizabeth University Hospital and the Royal Infirmary in Glasgow, and the Royal Alexandra Hospital in Paisley.

### Population
Patients who received hospital care for COVID-19, with or without admission, and were alive, were prospectively screened in real-time using an electronic healthcare information system (TrakCare®, InterSystems®, USA) and daily hospital reports identifying inpatients with laboratory-positive results for COVID-19.

The inclusion criteria were: (1) age ≥18 years old; (2) history of an unplanned hospital visit e.g., emergency department, or hospitalization >24 h for COVID-19 confirmed by a laboratory test (e.g., polymerase chain

reaction (PCR); (3) ability to comply with study procedures; and (4) ability to provide written informed consent.

The exclusion criteria were: (1) contra-indication to magnetic resonance (MR) imaging (e.g., severe claustrophobia, metallic foreign body); and (2) lack of informed consent.

### Ethics
The study obtained local institutional approval by NHS Greater Glasgow and Clyde Research and Development and.UK National Research Ethics Service (Reference 20/NS/0066). Participants gave informed consent to participate in the study before taking part.

### Statistics
The statistical analyses were pre-defined in a Statistical Analysis Plan. The statistical methods are described in the table legends.

### Reporting summary
Further information on research design is available in the Nature Portfolio Reporting Summary linked to this article.

## Results
In total, 1306 patients were screened between 22 May 2020 and 16 March 2021, and 267 provided written informed consent. The CONSORT flow diagram is provided in Fig. 1.

The SIMD quintile based on place of residence was available for 252 patients (Supplementary Data 1). One hundred and one (40%) patients were in the most deprived quintile (Q1), 56 (22%) in Q2, 31 (12%) in Q3, 23 (9%) in Q4 and 41 (16%) in the least deprived (Q5). Deprivation was negatively associated with attendance 28–60 days post-discharge ($p = 0.0378$).

### Clinical characteristics
The average age was 55 years, 227 (90%) were white, 101 (40%) were female, and 50 (20%) were healthcare workers; these characteristics were not associated with deprivation status (Supplementary Data 1).

People living in deprived areas tended to have higher body mass index (BMI, $p = 0.0444$), higher prevalence of diabetes mellitus ($p = 0.0239$), and higher Q-Risk 3 scores reflecting a greater likelihood of developing cardiovascular disease over the next 10 years ($p = 0.0124$).

### COVID-19 presentation
On admission to hospital (Supplementary Data 1), SIMD quintile was associated with the albumin plasma protein concentration ($p = 0.0435$), though with no clear trend, but not with acute phase reactants including ferritin ($p = 0.9489$), D-Dimer ($p = 0.6108$) or C-reactive protein ($p = 0.4137$). SIMD quintile was not associated with the WHO COVID-19 illness severity score ($p = 0.2781$) or the duration of admission ($p = 0.0643$).

### Cardiovascular phenotyping
One hundred and eighty-six (74%) patients reattended for investigation 28–60 days post-discharge (Supplementary Data 2). Attendance at 28–60 days post-discharge was associated with deprivation quintile ($p = 0.0378$), however, the clinical characteristics of the patients who did not attend were not significantly different compared to the patients who did attend (Supplementary Data 3).

Obstructive coronary artery disease ($p = 0.0128$) revealed by CT coronary angiography was associated with SIMD, being most prevalent in the most deprived quintile. Left ($p = 0.0044$) and right ($p < 0.0001$) ventricular end-diastolic volumes were highest in the least deprived quintile. No associations were seen in relation to myocardial inflammation ($p = 0.2972$), or scar ($p = 0.4946$) as revealed by cardiac MRI.

### Respiratory phenotyping
Patients living in areas of higher socioeconomic deprivation tended to show a higher prevalence of persistent ground glass opacity and/or consolidation

**Fig. 1 | Flow diagram of the clinical study.** The procedures involved screening hospitalised patients with COVID-19 defined by a PCR-positive result for SARS-CoV-2 in a nasopharyngeal swab and then obtaining written informed consent. The analysis population is defined by a PCR-positive result. Serial investigations were initiated in-hospital or early post-discharge (visit 1) and then repeated in association with multi-organ imaging at 28–60 days post-discharge (visit 2). Clinical follow-up continued for on average 450 days ± 88 s.d. (range, 290–627 days) post-discharge.

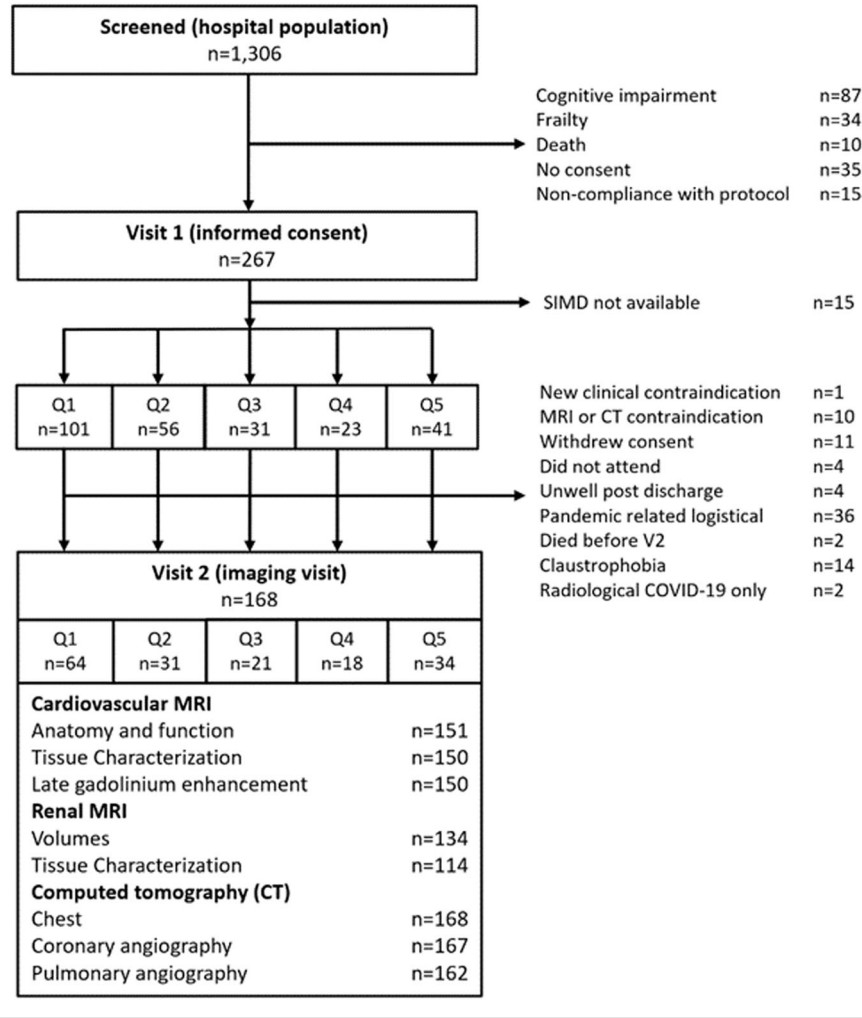

($p = 0.0085$) on CT Chest at 28–60 days, and, to a lesser extent, a higher estimated percentage of lung abnormality ($p = 0.0507$).

### Renal phenotyping
Deprivation was associated with renal inflammation, reflected by the average cortical T1 of the right and left kidneys at 28–60 days ($p = 0.0421$), though without any clear trend.

### Health-related quality of life, illness perception and physical function
Patient reported outcome measures were available for 184 (73%) patients (Supplementary Data 4). Increasing deprivation was associated with reported lower health-related quality of life (EQ-5D-5L utility score $p = 0.0084$), enhanced illness perception (BIPQ, $p = 0.0004$), and lower functional capacity (estimated peak oxygen consumption, $p = 0.0022$) at 28–60 days follow-up, but not at enrolment. These trends were driven by improvements between enrolment and 28–60 days amongst patients from less deprived areas, with little, or no, improvement amongst more deprived groups (Supplementary Data 5, 6, Fig. 2).

### Mental health
Overall mental health at enrolment varied between quintiles of deprivation (PHQ-4, $p = 0.0474$), though with no clear trend ($ptrend = 0.5041$). The association after 28–60 days was stronger ($p = 0.0038$) (Supplementary Data 4), with a trend towards less deprived groups showing better mental health ($ptrend = 0.0084$). This pattern appeared to be driven by depression scores improving over time amongst less deprived patients ($p = 0.0009$), whilst remaining stable in people living in deprived areas ($p = 0.4642$).

Whilst both PHQ-4 Anxiety and Depression scores demonstrated similar associations, this pattern was more evident for depression, with a significant interaction between time point and SIMD quintile ($p = 0.0203$). (Supplementary Data 5, Fig. 2).

### Clinical outcomes
Follow-up was continued to 13 December 2021 for all patients ($n = 252$, Supplementary Data 7). The mean (SD) duration of follow-up after hospital discharge for individuals included in this analysis was 428 (86) days (range, 290–627 days). Higher proportions of patients from deprived areas were referred to secondary care with symptoms consistent with NICE188 guideline[17] criteria for post-COVID-19 conditions (Long COVID), defined by the presence of symptoms at 28–84 days; $p = 0.0438$).

### Discussion
This study assessed the impact of socioeconomic status in an extensively phenotyped cohort of patients hospitalised with COVID-19 utilising serum and urine biochemistry, patient-reported outcomes and electrocardiograms at baseline and 28–60 days after discharge; multi-organ, cross-sectional imaging with computed tomography and magnetic resonance imaging and clinical follow-up up to a mean of 428 days after hospital discharge.

The distribution of socioeconomic status of the population was similar to that of the city of Glasgow and the surrounding area, with the majority (40%) of patients belonging to the most deprived quintile (Q1)[16]. As deprivation levels decrease, the proportion of patients in each subsequent quintile is reduced, with 22% in Q2, 12% in Q3, and 9% in Q4. Interestingly, there is a slight increase in the proportion of patients in the least deprived quintile (Q5) at 16%. People from more affluent areas are generally more

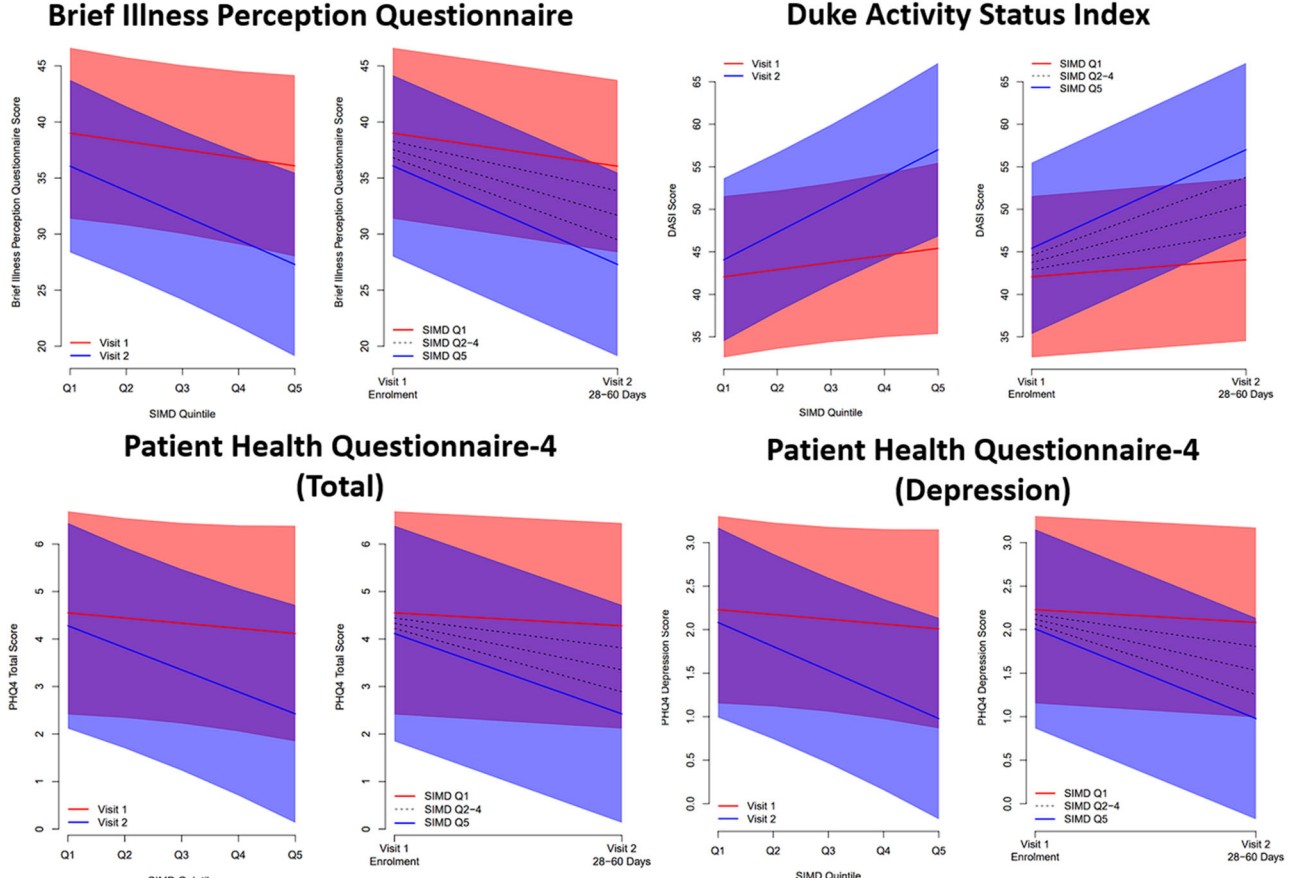

**Fig. 2 | Linear mixed effects regression models for patient reported outcomes in relation to SIMD.** Linear mixed effects regression models were fitted to assess the association between each PROM measure and deprivation (SIMD quintile) between enrolment and follow-up. Models included a random subject effect, with fixed effects for age (linear), sex (binary), SIMD quintile (linear), study visit (binary), and a SIMD-by-visit interaction. In each pair of figures, the left panel shows the predicted mean PROM score (with 95% confidence bands), in relation to SIMD quintile, at Visit 1 (enrolment) and Visit 2 (28–60 days post discharge). The right panel shows the predicted mean score in relation to study visit, for each SIMD quintile; confidence bands are shown for most and least deprived quintiles only. The data relating to this figure are available in Supplementary Data.

likely to participate in clinical trials, as such this could potentially be an example of Simpson's paradox meaning there are more of these individuals in our patient cohort even though their individual risk of hospitalisation is lower[18].

The main findings were, firstly, deprivation status was not associated with the initial severity of COVID-19 illness, yet it was associated with higher rates of persistent lung changes, lower health-related quality of life, enhanced illness perception, worse mental health, and lower functional capacity at 28–60 days follow-up. Secondly, patient reported outcomes showed little association with deprivation at enrolment. Whilst those from less deprived areas showed improvements over time, patients living in deprived areas in general did not, resulting in socioeconomic gradients by 28–60 days. Thirdly, higher proportions of patients from deprived areas were referred to secondary care with ongoing COVID-19 symptoms after discharge. Fourthly, patients from more deprived areas had increased cardiovascular risk scores at baseline and this correlated with the extent of coronary artery disease revealed by CT coronary angiography. Fifthly, patients from deprived areas had evidence of worse lifestyle circumstances with lower levels of baseline physical activity, higher mean body mass index and a higher prevalence of diabetes mellitus. Finally, patients from deprived areas had proportionately greater referral rates to secondary care, reflecting a greater burden of persisting health problems after COVID-19.

In the context of the medical literature, our findings concur with global observational data, highlighting a pervasive trend wherein lower socioeconomic status is associated with poorer outcomes following COVID-19 infection. This consistent pattern is observed across diverse geographical

regions and levels of affluence, underscoring the need for targeted interventions and support for vulnerable populations[19–24].

Our findings illustrate the adverse health implications of obesity and diabetes in individuals with COVID-19 illness[25]. In population studies, low socioeconomic status associates with a two-fold higher risk of having obesity[26], and BMI is associated with incomplete recovery after COVID-19[27]. Consumption of unhealthy energy-dense foods that are typically cheaper than healthy alternatives and reduced physical activity associate with deprivation status and obesity[28,29]. In our study, the mean plasma albumin concentration was inversely associated with deprivation status, consistent with worse nutrition.

Improving health disparities related to deprivation requires a multi-faceted approach that addresses the underlying social determinants of health.

Education and health literacy play a critical role in empowering individuals to make informed decisions about their health. Incorporating health education in school curricula, and providing accessible health information resources can help promote health literacy[30].

Economic policies that promote fair wages, job security, and social safety nets can reduce poverty and financial stress. These policies could include minimum wage increases, unemployment benefits, and affordable housing initiatives. Neighbourhood development programmes could be implemented to improve the physical environment, promote social cohesion, and increase access to essential services like healthy food, public transportation, and green spaces[31].

Data from the UK Biobank indicates that a mix of unhealthy lifestyle factors and disadvantaged socioeconomic status contribute to an elevated risk of severe COVID-19 outcomes[32]. The synergistic effect of these factors intensifies the probability of COVID-19 mortality and serious illness. While promoting healthier lifestyles can lower risks for everyone, focusing on support for socially deprived areas may result in more substantial public health improvements. Implementing public health campaigns and community-based programmes that endorse healthy behaviours, such as routine physical activity, balanced diets, and smoking cessation, is crucial for reducing health disparities.

Public health interventions may not mitigate genetic and environmental factors, and the role of individual choice for unhealthy behaviours[33]. Clinical trials are ongoing to assess whether weight loss interventions can help alleviate symptoms of Long COVID in people who are overweight[34].

Our data indicate socioeconomic differences in health outcomes develop within 28–60 days of index presentation. This time-period early after discharge from hospital presents an opportunity for preventive interventions e.g., dietary advice, physical rehabilitation, potentially targeting socially-deprived people who may be less well placed to engage with healthcare interventions.

To date, there are no evidence-based therapies for patients with persisting physical symptoms after COVID-19. To help address this gap, we developed a lifestyle intervention that may be helpful to patients with persisting symptoms in the recovery (or convalescence) phase over a 3-month period after COVID-19[35]. The intervention involves learning simple, resistance-based exercises, personalised according to the needs and circumstances of the individual who may be in-hospital or in the community. The rationale is to provide patients with a personalised, self-care, therapy option early during their recovery from COVID-19. In the current study, aerobic capacity after COVID-19, reflected by the DASI VO2 (ml/min) max at 28–60 days post-discharge, was inversely associated with deprivation status (Supplementary Data 4, 5). Therefore, physical rehabilitation may be helpful to patients from deprived circumstances after COVID-19. The programme does not require additional resources hence socioeconomic barriers to participation are minimised. The effects of this exercised-based intervention are being evaluated in a clinical trial (NCT04900961)[35].

Attendance at 28–60 days post-discharge was associated with deprivation quintile. This finding highlights that patients from socially deprived backgrounds engage less with medical follow-up. Rather than being a limitation of our study, this result highlights the need for targeting resources to facilitate access to healthcare for individuals from deprived backgrounds.

One quarter of the patients enrolled at baseline did not reattend at 28–60 days post-discharge and non-attendance was more prevalent in those from deprived backgrounds. The reasons included death and disability due to impairments in physical and cognitive function.

The sample size in CISCO-19 is consistent with a longitudinal study involving serial multi-organ assessments. However, to investigate the implications of our results on a population level, a larger study is required. Extending this analysis with a larger dataset such as the Lifelines Corona Study would allow us to assess the generalisability of our findings and to identify possible variations in the associations across different population groups and settings[36].

## Conclusions

In a post-hospital COVID-19 population, deprivation status associated with BMI, diabetes, coronary artery disease, impaired health status and may associate with Long COVID. Deprivation status influences illness trajectory after COVID.

## Data availability

Data requests will be considered by the Steering Group, which includes representatives of the Sponsor, the University of Glasgow, senior investigators independent of the research team, and the chief investigator. The Steering Group will take account of the scientific rationale, ethics, coordination, and resource implications. Data access requests should be submitted by email to the Chief Investigator (Colin Berry, corresponding author). The source data includes the deidentified numerical data used for the statistical analyses and deidentified imaging scans (MRI, CT) and ECGs. Data access will be provided through the secure analytical platform of the Robertson Centre for Biostatistics. This secure platform enables access to deidentified data for analytical purposes without the possibility of removing the data from the server. Requests for transfer of deidentified data (including source imaging scans) will be considered by the Steering Group, and if approved, a collaboration agreement would be expected. The Steering Group will consider any cost implications, and cost recovery would be expected on a not-for-profit basis. The source data underlying Fig. 2 including the deidentified numerical data can be found in Supplementary Data 6.

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

## Acknowledgements

We thank the staff and patients who are supporting this study, and the Chief Scientist Office of the Scottish Government for financial support. We thank the CISCO-19 Study Management Group (Mrs. Ammani Brown, Mrs. Chloe Cowan, Dr Lindsay Gillespie, Ms. Sharon Kean, Mr. Jurgen Van-Melckebeke, Dr Kim Moran-Jones, Dr Debra Stuart, and Dr Maureen Travers for their contributions towards the delivery of this study. CISCO-19 was an investigator-initiated clinical study that was funded by the Chief Scientist Office of the Scottish Government (COV/GLA/Portfolio project number 311300). The funder had no role in the design, conduct (non-voting TSC member), data analysis and interpretation, manuscript writing, or dissemination of the results. C.B., C.D., N.S. and R.M.T. were supported by the British Heart Foundation (RE/18/6134217). The MRI study involved technologies provided by Siemens Healthcare and the National Institutes of Health. HeartFlow (HeartFlow, Redwood City, CA) provided FFRCT. The study was co-sponsored by NHS Greater Glasgow & Clyde Health Board and the University of Glasgow.

## Author contributions

CB designed the study and wrote the first draft of the manuscript with AJM. AMcI and AMcC developed the statistical analysis plan and performed the statistical analyses. RS, MS, BZ, AK, CB, HKB, KGB, CC, VBG, LG, DG, PHB, AH, DJL, PWM, KJM, PBM, GR, NS, DS, SAS, RMT, SW, PW and KM contributed to either the design or delivery of the study. HB, DCa, DCo, IF, NNL, VL, RM, SN, AP, KER, NR, KR, GV, RPW and SWa served as expert members of the clinical events committee. All authors have given final approval for the current version to be published.

## Competing interests

C.B. is employed by the University of Glasgow, which holds consultancy and research agreements with Abbott Vascular, AstraZeneca, Boehringer Ingelheim, Coroventis, GSK, HeartFlow, Menarini, Novartis, Siemens Healthcare, Somalogic and Valo Health. These companies had no role in the design or conduct of the study, or in the data collection, interpretation, or reporting. HeartFlow derived FFRCT. None of the other authors have any relevant disclosures.

## Additional information

Andrew J. Morrow[1,2], Robert Sykes[1,2], Merna Saleh[2], Baryab Zahra[2], Alasdair MacIntosh[3], Anna Kamdar[1], Catherine Bagot[4], Hannah K. Bayes[5], Kevin G. Blyth[6,7], Heerajnarain Bulluck[8], David Carrick[9], Colin Church[6,10], David Corcoran[1,2], Iain Findlay[11], Vivienne B. Gibson[4], Lynsey Gillespie[12], Douglas Grieve[13], Pauline Hall Barrientos[14], Antonia Ho[15], Ninian N. Lang[1,2], David J. Lowe[16], Vera Lennie[17], Peter W. Macfarlane[18], Kaitlin J. Mayne[1,19], Patrick B. Mark[1,19], Alex McConnachie[3], Ross McGeoch[10], Sabrina Nordin[2], Alexander Payne[20], Alastair J. Rankin[1], Keith Robertson[11], Nicola Ryan[17], Giles Roditi[21], Naveed Sattar[1], David Stobo[21], Sarah Allwood-Spiers[5], Rhian M. Touyz[1], Gruschen Veldtman[22], Sarah Weeden[3], Robin Weir[10], Stuart Watkins[11], Paul Welsh[1], Kenneth Mangion[1,2] & Colin Berry[1,2] ✉

[1]School of Cardiovascular and Metabolic Health, University of Glasgow, Glasgow, UK. [2]Department of Cardiology, Queen Elizabeth University Hospital, Glasgow, UK. [3]Robertson Centre for Biostatistics, University of Glasgow, Glasgow, UK. [4]Department of Haemostasis and Thrombosis, Glasgow Royal Infirmary, Glasgow, UK. [5]Department of Respiratory Medicine, Glasgow Royal Infirmary, Glasgow, UK. [6]Department of Respiratory Medicine, Queen Elizabeth University Hospital, Glasgow, UK. [7]Institute of Cancer Sciences, University of Glasgow, Glasgow, UK. [8]Leeds General Infirmary, St James's University Hospital, Leeds, UK. [9]Department of Cardiology, University Hospital Hairmyres, East Kilbride, UK. [10]Regional Heart and Lung Centre, NHS Golden Jubilee, Clydebank, UK. [11]Department of Cardiology, Royal Alexandra Hospital, Paisley, UK. [12]Project Management Unit, Glasgow Clinical Research Facility, Greater Glasgow and Clyde Health Board, Glasgow, UK. [13]Department of Respiratory Medicine, Royal Alexandra Hospital, Glasgow, UK. [14]Department of Medical Physics, NHS Greater Glasgow and Clyde Health Board, Glasgow, UK. [15]MRC-University of Glasgow Centre for Virus Research, Glasgow, UK. [16]Department of Emergency Medicine, Queen Elizabeth University Hospital, NHS Greater Glasgow and Clyde Health Board, Glasgow, UK. [17]Department of Cardiology, Aberdeen Royal Infirmary, Aberdeen, UK. [18]Electrocardiology Core Laboratory, Institute of Health and Wellbeing, University of Glasgow, Glasgow, UK. [19]Glasgow Renal and Transplant Unit, Queen Elizabeth University Hospital, NHS Greater Glasgow and Clyde Health Board, Glasgow, UK. [20]Department of Cardiology, University Hospital Crosshouse, Kilmarnock, UK. [21]Department of Radiology, NHS Greater Glasgow and Clyde Health Board, Glasgow, UK. [22]Scottish Adult Congenital Cardiac Service, NHS Golden Jubilee, Clydebank, UK. ✉e-mail: colin.berry@glasgow.ac.uk

