## [Peer Review File · Communications Medicine]

Socioeconomic deprivation and illness trajectory in the Scottish population after COVID-19 hospitalizationReferee expertise:

Referee #1: Public health, publications on socioeconomic status, COVID-19, modeling approaches

Referee #2: MD in cardiology and clinical data scientist, publications on long-COVID

Reviewers' comments:

Reviewer #1 (Remarks to the Author):

This is an interesting, well-executed paper that adds to the growing evidence that the covid-19 pandemic has had a differential impact across socioeconomic groups. The consensus that is starting to emerge is that individuals with higher level of neighborhood and/or individual socioeconomic deprivation had a higher risk of contracting covid, conditional on contracting it had a higher risk of hospitalization and conditional on hospitalization worse outcomes. This paper adds to the latter of these relationships, thus highlighting that deprivation affects health at each point of the disease cycle.

The particular strength of this paper is that it highlights how various specific disease areas are affected by deprivation after covid hospitalization.

Some things to consider would be:

- 1.) Are the elevated disease patterns consistent with Long-Covid symptoms or these separate from that? That is, do we interpret this paper as highlighting that long-covid is more prevalent among individuals with a high degree of neighborhood deprivation?
- 2.) Is there an interaction between individual and neighborhood deprivation? Recent evidence has highlighted that such interactions exist for other disease patterns (see, for instance, <https://academic.oup.com/ije/article/50/6/1959/6248208>).
- 3.) What are the implications on a population level? The CISCO-19 study is very good in the clinical setting but to get an idea at the population level a larger study would be required. It would be interesting to extend the analysis in follow-up study using, for instance, the Lifelines Corona Study (<https://bmjopen.bmj.com/content/11/3/e044474.abstract>) in which also data on hospitalization was collected.
- 4.) I thought that the gradient of covid presentation at the hospital (line 147 - 150) was worth some more discussion. From Q1 to Q4 it is as expected (Q1 highest percentage of presentation, Q4 lowest) but then there is a jump at Q5. Why is that? Are there large Q5 areas around so that by Simpsons Paradox there are more of them in the hospitalization group even though their individual risk of hospitalization is lower?
- 5.) I found the extensive discussion of the smoking related public health interventions somewhat laborious and not necessarily on point. My interpretation would be that the interventions should not take place at the public health level but really intervene at the level of deprivation. Income support, neighborhood development, etc.

Reviewer #2 (Remarks to the Author):

This is an important post-COVID cohort study which focuses on the role of socioeconomic status. Although there have been many studies from the UK context, there are relatively few from Scotland

and so these are novel data.

The background, methods and results are appropriate and rigorous.

My only minor suggestions are:

1. to reference international literature for comparison in the discussion
2. to incorporate the intersection of other risk factors with SES in the discussion:<https://pubmed.ncbi.nlm.nih.gov/35351028/>

Reviewer #1:

1. **Are the elevated disease patterns consistent with Long-Covid symptoms or these separate from that? That is, do we interpret this paper as highlighting that long-covid is more prevalent among individuals with a high degree of neighborhood deprivation?**

RESPONSE: Our results indicate that a greater percentage of patients from deprived areas were referred to secondary care for ongoing COVID-19 symptoms (aligning with the NICE188 guideline criteria for long COVID) following discharge. This suggests that Long COVID may be more common among individuals residing in neighbourhoods with higher levels of deprivation.

The blunted improvement in health-related quality of life, illness perception, and physical function observed between enrolment and follow-up, as demonstrated in Table 4 and Figure 2, further substantiates the presence of ongoing symptoms.

A short statement to this effect has been added to the conclusion for clarity (Page 15, Line 299):

“In a post-hospital COVID-19 population, deprivation status associated with BMI, diabetes, coronary artery disease, impaired health status and may associate with Long COVID. Deprivation status influences illness trajectory after COVID.”

2. **Is there an interaction between individual and neighborhood deprivation? Recent evidence has highlighted that such interactions exist for other disease patterns (see, for instance, <https://academic.oup.com/ije/article/50/6/1959/6248208>).**

RESPONSE: Thank you for this suggestion and insightful reference. Unfortunately, we did not collect data on individual participant income, education or employment status. As such we are unable to provide an accurate composite of health-related lifestyle risk factors (lifestyle risk index). We intend to utilise the iPCQ questionnaire in future studies so that we may be able to contribute to this important question.

3. **What are the implications on a population level? The CISCO-19 study is very good in the clinical setting but to get an idea at the population level a larger study would be required. It would be interesting to extend the analysis in follow-up study using, for instance, the Lifelines Corona Study (<https://bmjopen.bmj.com/content/11/3/e044474.abstract>) in which also data on hospitalization was collected.**

RESPONSE: We appreciate your valuable input regarding the potential implications of our study at the population level and the suggestion to extend our analysis using a larger dataset, such as the Lifelines Corona Study.

The Lifelines Corona Study offers a valuable opportunity to assess the relationship between deprivation and disease outcomes at the population level over a similar time frame as CISCO-19. This would allow us to assess the generalisability of our findings and to identify possible variations in the associations across different population groups and settings.

In light of your suggestion, we have added a section to the Limitations section of our discussion (Page 15, Lines 293-296):

“The sample size in CISCO-19 is consistent with a longitudinal study involving serial multi-organ assessments. However, to investigate the implications of our results on a population level, a larger study is required. Extending this analysis with a larger dataset such as the Lifelines Corona Study would allow us to assess the generalisability of our findings and to identify possible variations in the associations across different population groups and settings. [34]”

- 4. I thought that the gradient of covid presentation at the hospital (line 147 - 150) was worth some more discussion. From Q1 to Q4 it is a expected (Q1 highest percentage of presentation, Q4 lowest) but then there is a jump at Q5. Why is that? Are there large Q5 areas around so that by Simpsons Paradox there are more of them in the hospitalization group even though their individual risk of hospitalization is lower?**

RESPONSE: Thank you for your insightful comment regarding the gradient of COVID-19 presentation at the hospital across different deprivation quintiles (Q1 to Q5). We agree that the observed pattern, with a jump in presentation at Q5, warrants further discussion. One possible explanation for this unexpected pattern, as you suggested, could be the presence of Simpson's Paradox. We have added a section to the Discussion to reflect this (Page 11, Lines 209-216).

“The distribution of socioeconomic status of the population was similar to that of the city of Glasgow and the surrounding area, with the majority (40%) of patients belonging to the most deprived quintile (Q1). [14] As deprivation levels decrease, the proportion of patients in each subsequent quintile is reduced, with 22% in Q2, 12% in Q3, and 9% in Q4. Interestingly, there is a slight increase in the proportion of patients in the least deprived quintile (Q5) at 16%. People from more affluent areas are generally more likely to participate in clinical trials, as such this could potentially be an example of

Simpson's paradox meaning there are more of these individuals in our patient cohort even though their individual risk of hospitalisation is lower.[16]”

- 5. I found the extensive discussion of the smoking related public health interventions somewhat laborious and not necessarily on point. My interpretation would be that the interventions should not take place at the public health level but really intervene at the level of deprivation. Income support, neighborhood development, etc.**

RESPONSE: We appreciate your feedback regarding the discussion of smoking-related public health interventions. We understand that the focus on these interventions may have been too narrow and not entirely relevant to the broader issue of addressing deprivation and its impact on COVID-19 outcomes.

In light of your comment, we have revised the discussion to better emphasise the importance of targeting the root causes of deprivation. This section now reads (Page 13, line 244-266):

“Improving health disparities related to deprivation requires a multifaceted approach that addresses the underlying social determinants of health.

Education and health literacy play a critical role in empowering individuals to make informed decisions about their health. Incorporating health education in school curricula, and providing accessible health information resources can help promote health literacy. [28]

Economic policies that promote fair wages, job security, and social safety nets can reduce poverty and financial stress. These policies could include minimum wage increases, unemployment benefits, and affordable housing initiatives. Neighbourhood development programs could be implemented to improve the physical environment, promote social cohesion, and increase access to essential services like healthy food, public transportation, and green spaces. [29]

Data from the UK Biobank indicates that a mix of unhealthy lifestyle factors and disadvantaged socioeconomic status contribute to an elevated risk of severe COVID-19 outcomes. [30] The synergistic effect of these factors intensifies the probability of COVID-19 mortality and serious illness.

While promoting healthier lifestyles can lower risks for everyone, focusing on support for socially deprived areas may result in more substantial public health improvements. Implementing public health campaigns and community-based programs that endorse healthy behaviours, such as routine physical activity, balanced diets, and smoking cessation, is crucial for reducing health disparities.

Public health interventions may not mitigate genetic and environmental factors, and the role of individual choice for unhealthy behaviours. [31] Clinical trials are ongoing to assess whether weight loss interventions can help alleviate symptoms of Long COVID in people who are overweight. [32]”

Reviewer #2:

- 1. This is an important post-COVID cohort study which focuses on the role of socioeconomic status. Although there have been many studies from the UK context, there are relatively few from Scotland and so these are novel data. The background, methods and results are appropriate and rigorous.**

Response: Thank you for your positive feedback and recognition of the novelty of our study focusing on the role of socioeconomic status in the context of Scotland.

- 2. To reference international literature for comparison in the discussion**

Response: Thank you for this suggestion, we have added the following paragraph that refers to additional studies from each continent, covering a range of wide range of affluence (France, USA, Chile, Nigeria and China): (Page 12, Lines 231-235)

“Our findings concur with global observational data, highlighting a pervasive trend wherein lower socioeconomic status is associated with poorer outcomes following COVID-19 infection. This consistent pattern is observed across diverse geographical regions and levels of affluence, underscoring the need for targeted interventions and support for vulnerable populations.”[17–22]

3. To incorporate the intersection of other risk factors with SES in the discussion:<https://pubmed.ncbi.nlm.nih.gov/35351028/>

Response: Thank you for this suggestion. In response to helpful comments from both reviewers we have re-written the Discussion, providing more detail on the determinants of socioeconomic status. In response to this comment we added a section discussing the excellent suggested reference and the potential impact of targeted public health interventions. (Page 13, lines 255-262)

“Data from the UK Biobank indicate that a mix of unhealthy lifestyle factors and disadvantaged socioeconomic status contribute to an elevated risk of severe COVID-19 outcomes. [30] The synergistic effect of these factors increases the likelihood of COVID-19 mortality and serious illness. While promoting healthier lifestyles can lower risks for everyone, focusing on support for socially deprived areas may result in more substantial public health improvements. Implementing public health campaigns and community-based programs that endorse healthy behaviours, such as routine physical activity, balanced diets, and smoking cessation, is crucial for reducing health disparities.”

The authors are very grateful the constructive feedback and opportunity to resubmit an improved manuscript that we hope you find suitable for publication.

REVIEWERS' COMMENTS:

Reviewer #1 (Remarks to the Author):

Thank you for engaging with my comments and suggestions. I am happy with how the paper has turned out now.